# Trajectories of Depressive Symptoms and Anxiety during Pregnancy and Associations with Pregnancy Stress

**DOI:** 10.3390/ijerph18052733

**Published:** 2021-03-08

**Authors:** Hyejung Lee, Ki-Eun Kim, Mi-Young Kim, Chang Gi Park, Jung Yeol Han, Eun Jeong Choi

**Affiliations:** 1Mo-Im Kim Nursing Research Institute, Yonsei University College of Nursing, Seoul 03722, Korea; hlee26@yuhs.ac; 2College of Nursing, Yonsei University, Seoul 03722, Korea; 3College of Nursing, Woosuk University, Wanju-gun, Jeollabuk-do 55338, Korea; miyoungkim726@gmail.com; 4College of Nursing, University of Illinois at Chicago, Chicago, IL 60612, USA; parkcg@uic.edu; 5Department of Obstetrics and Gynecology, Inje University Ilsan Paik Hospital, Goyang 10380, Korea; hanjungyeol055@gmail.com (J.Y.H.); spica0924@gmail.com (E.J.C.)

**Keywords:** depression, anxiety, cortisol, pregnancy, trajectory analysis, Korea

## Abstract

The purposes of this study were to investigate the trajectory groups of depressive symptoms and anxiety in women during pregnancy and to identify the factors associated with those groups. Participants were recruited from the outpatient clinic of a women’s health hospital in Seoul, Korea. Pregnant women (*n* = 136) completed a survey questionnaire that included questions on depressive symptoms, anxiety, and pregnancy stress; additionally, their saliva was tested for cortisol hormone levels three times during their pregnancies. The group-based trajectory modeling approach was used to identify latent trajectory groups. Ordinal logistic regressions were used to explore the association of latent trajectory groups with sociodemographic factors and pregnancy stress. Three trajectory groups of depressive symptoms were identified: low-stable (70%), moderate-stable (25%), and increased (5%). Four trajectory groups of anxiety were identified: very low-stable (10%), low-stable (67%), moderate-stable (18%), and high-stable (5%). The only factor associated with both the depressive symptoms and anxiety trajectory groups was pregnancy stress (*p* < 0.001). Most participants showed stable emotional status; however, some participants experienced higher levels of depressive symptoms and anxiety related to higher pregnancy stress. These pregnant women may need additional care from healthcare providers to promote their wellbeing during pregnancy.

## 1. Introduction

Mental health problems such as depression and anxiety are prevalent among pregnant women, with rates as high as 30% [1,2,3]. The mental health of pregnant women affects the health outcomes of both the pregnant women and their infants [4]. Infants born to mothers who have experienced severe perinatal stress are reported to have altered behavioral responses to stress. Changes in the antenatal stress-related biology of pregnant women may be linked to adverse behavioral responses toward their infants [5]. In addition, depression during pregnancy is often associated with adverse birth outcomes, including preterm birth, low infant birth weight, and fetal growth restriction [2,6]. Pregnancy is a time of increased vulnerability to the development of depression. Prevalence of depression has been estimated at 11.9% among women during the perinatal period, and there is a significantly higher prevalence in low- to middle-income countries compared to high-income countries [7]. In Korea, between 20% and 40% of prenatal women have been reported to have depression, as measured by varying scales [8].

Anxiety is another prevalent problem faced by pregnant women [1,9]. As their due dates approach, pregnant women are likely to express anxiety and fear regarding the labor and delivery process [10]. However, multiparous women who have experienced successful labor and delivery may have lower anxiety levels than primipara women [11]. Similarly, certain subgroups of pregnant women may be more vulnerable to anxiety. A strong relationship between anxiety and depression has been established among pregnant women [12]. Three out of every four anxious women are reported to suffer from comorbid depressive symptoms. In addition, several studies have reported that both anxiety and depressive symptoms follow a U-shaped curve, with the highest levels of these symptoms occurring during the first and third trimesters of pregnancy [9,13]. Given the high correlation of anxiety and depression, it is essential to identify the trajectories of anxiety and depressive symptoms simultaneously during pregnancy. This type of approach would help to explore the risk for negative mental health and to provide care for pregnant women during the appropriate time window. Various factors influencing prenatal anxiety and depression in women have been investigated; however, inconsistent reports are available regarding stress, sleep quality, and daily physical activity. Mental health problems before pregnancy, on the other hand, have been identified as strong predictors of perinatal and postnatal mental health problems [14,15,16,17]. Yet, modifiable influencing factors need to be identified to prevent the negative consequences of untreated prenatal anxiety and depression on both mothers and infants [2,18].

In humans, cortisol helps the body cope with stressful situations, but prolonged exposure to elevated levels of stress causes severe fatigue, anxiety, and depression [4]. As a biomarker of stress, cortisol measurements have been widely utilized in clinics and research [19]. Cortisol levels in the blood are closely correlated with stress levels in healthy populations and pregnant women [5,20]. Offering a non-invasive measure, cortisol levels in saliva are also related to anxiety and depression in pregnant women [19,21,22].

Although there has been extensive research on mental health during pregnancy, it is still necessary to investigate pregnant women’s patterns of mental health changes as gestation progresses, as well as the underlying mechanisms; this must be approached using objective measures, such as biomarkers. Thus, we added a saliva cortisol measure to the psychobiological variables of women during pregnancy to determine the differences between the distinct groups identified.

Thus, the primary objective of this study was to investigate the trajectory groups of depressive symptoms and anxiety of women during pregnancy, and the secondary objective was to identify the factors associated with those groups.

## 2. Materials and Methods

### 2.1. Study Design and Participants

This study used a repeated measures design with a convenience sampling method. After approval of the study protocol by the Institutional Review Board was obtained, data collection began. The participants were recruited from the outpatient clinic of a women’s health hospital located in Seoul, Korea. Pregnant women were eligible for participation if they were over 20 years of age, their pregnancy (<16 weeks of gestation) was confirmed by a medical doctor, and they had no known medical illnesses. Exclusion criteria were twin or triplet pregnancy and diagnosed pregnancy complications such as gestational hypertension or gestational diabetes mellitus. During the examination, an obstetric physician briefly introduced the study to each potential participant; a research assistant provided the detailed research protocol and obtained consent from participants who showed interest in participation in the study. The survey questionnaire included questions addressing depression, anxiety, pregnancy-related stress, and sociodemographic characteristics. Participants were asked to complete the survey questionnaire during the regular prenatal check-up for each trimester (a total of three times). Of the 149 women who initially participated, those who reported an abortion (*n* = 3), transferred to another hospital (*n* = 6), or provided incomplete surveys (*n* = 4) were excluded. Therefore, the final dataset included 136 surveys that were suitable for statistical analysis.

### 2.2. Measures

Before data collection began, trained research assistants provided a detailed research protocol to potential participants and obtained their informed consent. Participants were asked to complete the questionnaire and provide their saliva in a clean tube to a research assistant. The tube with saliva was immediately stored in an icebox and delivered to the lab at the college of nursing where the analysis was performed. For the second- and third-trimester data collections, appointments with the women’s obstetric physicians were recorded, and the same data collection method was used.

#### 2.2.1. Depressive Symptoms

Depressive symptoms were measured using the Korean version of the Center for Epidemiologic Studies Depression scale (CES-D), whose validity and reliability have been confirmed within the Korean population. The scale assesses depressive symptoms over the past week and consists of 20 items scored on a 4-point Likert scale (ranging from 0 to 3). Total scores range from 0 to 60, with higher scores indicating more depressive symptoms. Scores above 16 are known to indicate severe depressive symptoms [23]. Cronbach’s α was 0.86 in the present study.

#### 2.2.2. Anxiety

Anxiety was measured using the Korean version of Spielberg’s State-Trait Anxiety Inventory (STAI-S), which includes 20 items scored on a 4-point Likert scale. Total scores range from 20 to 80, with higher scores indicating a higher state of anxiety. The optimum cut-off for the STAI-S during pregnancy is 40 (anxious vs. non-anxious women) [24]. Cronbach’s α of the STAI-S in the present study was 0.92.

#### 2.2.3. Pregnancy Stress

Pregnancy stress was measured using the pregnancy-related stress scale for pregnant Korean women, developed by Ahn [25] and revised and validated by Lee and Seo [26]. This scale includes 20 items scored on a 5-point Likert scale. Total scores range from 20 to 100, with higher scores indicating greater stress related to pregnancy. Cronbach’s α was 0.89 in the revised study [26] and 0.88 in the present study.

#### 2.2.4. Cortisol Level

In addition to the psychobiological variables, we included the saliva cortisol levels of the women during early pregnancy to determine the differences between the distinct groups identified. Given the diurnal characteristics of human cortisol levels and that the quality of cortisol samples is affected by sampling time, morning is considered the most common time of day for cortisol collection [21,27]. Therefore, we collected the saliva in the morning, between 10:00 a.m. and 12:00 p.m., at the outpatient clinic. Before participants provided saliva samples, they washed their mouths with clean water. The saliva samples were transported on ice bags and stored at −20 °C in the nursing college’s lab until the analysis was conducted. Salivary cortisol levels were measured using enzyme-linked immunosorbent assay kits supplied commercially for single measurements; the supplied standards were assayed in duplicates (R&D Systems, Minneapolis, MN, USA). The intra- and inter-assay coefficients of variation for salivary cortisol were 6.9% ± 1.9% and 9.9% ± 0.8%, respectively, which are both within recommended ranges [28].

Cortisol levels generally show skewed distributions within healthy populations. As changes in cortisol levels related to stress are known to be non-linear, it is appropriate to compare pregnant women with high levels of cortisol to those with low levels, rather than using cortisol levels as continuous variables. Thus, in the data analysis, comparison was made between one group containing three-quarters of the total cortisol level distribution and another group containing the remaining quarter. When a cortisol level was within the lower three quantiles, it was assigned 0; when it was within the highest quantile, it was assigned 1.

#### 2.2.5. Demographic Information

Information regarding each participant’s age, height, weight, education level, current employment status, annual household income (very low, low, middle, and high), planned pregnancy, and gestational age were obtained. Age was categorized as <35 or ≥35 years. Each participant’s body mass index (BMI) was computed and categorized as underweight (<18.5 kg/m^2^), healthy weight (18–23 kg/m^2^), overweight (23–25 kg/m^2^), or obese (≥25 kg/m^2^) based on the Asian criteria.

### 2.3. Statistical Analysis

Statistical analysis was performed using IBM SPSS 25 (IBM Corp, Chicago, IL, USA) and Stata 16 (Stata Corp LLC, College Station, TX, USA). Descriptive statistics were used to analyze the mean, standard deviation, and frequency of characteristics.

The group-based trajectory modeling approach was used to identify the trajectory groups of depressive symptoms based on the total CES-D scores and anxiety based on the STAI-S scores from early to late pregnancy [29]. A series of trajectory model specifications were systematically tested by determining whether shapes of depressive symptom and anxiety trajectories were linear, quadratic, or cubic according to the Bayesian information criterion (BIC). The number of trajectory groups in the model was increased by 1 and repeated until the best-fit model was found. Once the best-fit model was identified, a posteriori analysis was performed; the demographic and individual variables were compared between the trajectory groups with the Kruskal-Wallis test and Fisher’s exact test. This was performed because the data were unequally distributed among the cells of the contingency table.

The ordinal logistic regression model was used to identify factors associated with depressive symptoms and anxiety in pregnant women. We checked the parallel regression assumption to ensure the coefficients were in the same direction for each category [30]. Although the ordinal logistic regression on anxiety trajectory groups violated the parallel assumption, gologit2 with the autofit command confirmed the parallel assumption; there was no difference in the estimated coefficient values. Finally, the ordinal logistic regressions were conducted with the simultaneous entry of sociodemographic covariates. Adjusted odds ratios (ORs) and 95% confidence intervals (CIs) are presented to reflect associations of strength and significance.

## 3. Results

### 3.1. Characteristics of Participants

The characteristics of participants by trimester are summarized in Table 1. Over 30% of the participants’ ages were over 35 years, and most (90%) had degrees beyond the college level. More than half of the participants (55%) reported a planned pregnancy, and about half of the participants reported middle-class household incomes. Two-thirds of the participants were employed during pregnancy. About three-fifths of the participants (59%) reported a healthy pre-pregnancy BMI. Almost none of the participants consumed alcohol or smoked cigarettes during their pregnancies (alcohol, ≤10%; smoke, ≤1%). Mean cortisol levels increased as the trimesters progressed. However, pregnancy stress increased until the second trimester and stabilized thereafter.

### 3.2. Correlation between Depressive Symptoms, Anxiety, and Pregnancy Stress in Each Trimester

Associations between depressive symptoms, anxiety, and pregnancy-related stress of pregnant women are statistically significant in each trimester. All *p*-values of correlation are smaller than 0.01, as illustrated in Table 2. In all trimesters, depressive symptoms and stress were highly correlated, and anxiety and stress were also significantly correlated.

### 3.3. Description of Trajectory Group Patterns

A series of model specifications were systematically tested by changing the order of group numbers and trajectory polynomials (linear, quadratic, and cubic) to choose the best-fit model using the minimum BIC [29]. Since we conducted measurements three times during the pregnancies, a curve wave was not appropriate.

Based on the longitudinal patterns of depressive symptoms and the most optimized for fit with the minimum BIC and quadratic trajectory polynomial, three latent trajectory groups were identified (Figure 1). These groups were descriptively labeled according to their patterns of depressive symptoms: (1) low-stable, (2) moderate-stable, and (3) increased. In this model, the “low-stable” group, which is the largest trajectory group, includes pregnant women who consistently showed low-stable depressive symptoms and accounts for 70% of the participants (*n* = 96). The second largest group (25%), the “moderate-stable” trajectory group, includes women who showed suboptimal depressive symptoms that remained consistent. The last trajectory group, “increased” (5%), includes women whose scores were higher than 16 and whose scores increased in the second trimester and remained the same during the last trimester.

Based on the longitudinal patterns of anxiety and the most optimized for fit with the minimum BIC and linear trajectory polynomial, four latent trajectory groups were identified (Figure 2). These were descriptively labeled according to their patterns of anxiety: (1) very low-stable, (2) low-stable (3) moderate-stable, and (4) high-stable. The first trajectory group, “very low-stable” (10%), is composed of women whose anxiety scores were the lowest of the participants and relatively stable throughout pregnancy. The largest trajectory group (67%) was the “low-stable” group, and these anxiety scores were constant at a higher level than those of the “very low-stable” group. The third trajectory group, “moderate-stable,” includes women who had moderate-to-high anxiety scores throughout pregnancy; approximately one-fifth of the participants (18%) were in this group. The last and smallest trajectory group, “high-stable,” includes women who had the highest anxiety scores throughout pregnancy.

#### 3.3.1. Sociodemographic Characteristics by Latent Trajectory Groups of Depressive Symptoms and Anxiety

Table 3 presents the characteristics of the latent trajectory groups for depressive symptoms and anxiety. Pregnancy stress was significantly different in the trajectory groups of both depressive symptoms and anxiety (*p* < 0.001 and *p* = 0.017, respectively. 

Compared to the low-stable depressive symptoms group, the moderate and increased groups showed significantly higher pregnancy stress (*p* < 0.001 and *p* = 0.038, respectively). The very low-stable anxiety group showed significantly higher pregnancy stress than the moderate-stable anxiety group (*p* = 0.029). No other characteristics were found to differ among the trajectory groups for depressive symptoms and anxiety.

#### 3.3.2. Factors Associated with Trajectory Groups of Depressive Symptoms and Anxiety

Table 4 presents the factors associated with the trajectory groups of depressive symptoms and anxiety. Pregnancy stress significantly differed among the trajectory groups of depressive symptoms (*p* < 0.01). For a one-unit increase in pregnancy stress, the odds of a higher category group are 1.13 times greater in depressive symptoms (for example, the low-stable group [group 1] versus the moderate-stable group [group 2]) when other variables in the model are held constant. The same amount, which is 1.13 times greater for depressive symptoms, can be also applied between the moderate-stable and increased groups.

In terms of anxiety, pregnancy stress significantly differed among the trajectory groups of anxiety (*p* < 0.01). For a one-unit increase in pregnancy stress, the odds of a higher category group are 1.09 times greater in anxiety (for example, the very-low stable group [group 1] versus the low stable group [group 2]) when the other variables in the model remain constant. The same increase, 1.09 times, can be applied between the low-stable group (group 2) and the moderate-stable group (group 3).

## 4. Discussion

This longitudinal study identified distinct trajectory groups of depressive symptoms and anxiety levels in Korean women during pregnancy and explored potentially influencing factors. Three latent trajectory groups of depressive symptoms and four latent trajectory groups of anxiety were identified. The three latent trajectory groups were ‘low-stable’ (70%), ‘moderate-stable’ (25%), and ‘increased’ (5%). The four anxiety trajectory groups were very low-stable (10%), low-stable (67%), moderate-stable (18%), and high-stable (5%).

Depression and anxiety are common complaints among pregnant women and are sometimes severe for certain women during the prenatal period [10,31]. The trajectory groups based on patterns of depressive symptoms identified three heterogeneous groups: low, moderate, and increased depressive symptom levels. This finding is consistent with previous studies demonstrating the “low” trajectory represented by a stable pattern of low symptom levels over time, which usually accounts for the largest trajectory group (70%) [32]. The “increased” trajectory group also aligned with previous studies’ results; this group had increased depressive symptoms in the second trimester that persisted over time, forming the smallest trajectory group (≤10%) [32]. Despite being the smallest group in this study, this group of women should be investigated as an at-risk group for mental health problems because all women in the “increased” trajectory group scored over 16 on the CES-D, reflecting high levels of depressive symptoms. Tailored and continuous intervention for this group is required during early pregnancy.

Considering the four anxiety trajectory groups, our results are consistent with previous studies [1,33] showing that the trajectory patterns of anxiety were stable. Anxiety during pregnancy seems to remain consistent without abrupt changes, although state anxiety (rather than trait anxiety) was measured in this study. Being pregnant involves daily life that differs from the pre-pregnancy period, requiring caution against excessive weight gain and the need to maintain a good diet and sleep behavior. This finding also implies that an initial assessment of anxiety during early pregnancy can be used predictively, and high anxiety levels need management accordingly, due to their persistence throughout pregnancy [1].

Stress related to pregnancy is differentiated among the trajectory groups of both depressive symptoms and anxiety, according to the results from both the univariate and ordinal logistic regression analyses in this study. Stress is known to significantly and negatively impact the mental health of pregnant women. Perceived stress was associated with high levels of anxiety early in pregnancy [34]. Pregnancy-related stress can be related to negative emotions and cognitions about pregnancy and birth. Although a longitudinal design was used, the causal relationship between stress and depressive symptoms and anxiety cannot be identified.

Several studies have reported significant associations between antenatal depression and anxiety and sociodemographic characteristics, including age, education, and income [10,32,35], but no significant characteristics were noted in the present study. The majority of participants in this study were mature and older, ranging from 25 to 42 years, and more than 90% were at least college graduates. Almost none of the participants smoked cigarettes or consumed alcohol during their current pregnancies. These characteristics demonstrate that they were homogeneous and higher in socioeconomic status than the general population, which may affect mental health status. Past psychiatric history, post-traumatic stress disorder, and partner (or family) relationships were found to adversely affect the mental health of pregnant women [10].

This study is unique as it included cortisol levels to identify factors associated with mental health in pregnant women. Similar to previous studies, the average cortisol levels were increased during pregnancy [36]. However, the cortisol levels were not found to be a significant factor in the present study. This may be due to the fact that cortisol was measured using saliva that was only collected once, thus potentially not offering an accurate reflection of the participants’ overall stress levels. In fact, a few previous studies reported a lack of concordance in subjective (questionnaires) and objective (salivary cortisol) stress measures [37,38].

Considering the underdiagnosis of depression and anxiety during pregnancy, the results of this study are extremely important. Stress related to pregnancy itself during this period, as the only factor affecting mental health, should be assessed due to its interrelations with other aspects of mental health. The major finding of this study indicates that a small number of women showed increased depressive symptoms as their pregnancies progressed; therefore, greater attention and care are needed to prevent adverse obstetric outcomes within this group of women. Unlike depressive symptoms, anxiety assessed during early pregnancy did not change abruptly, so early and timely screening by healthcare providers could reduce anxiety symptoms in pregnant women. Rather than dismissing the efforts to reduce symptoms of depression and anxiety as a matter for individual patients, healthcare providers should assess for and provide information about stress, depression, and anxiety in places such as obstetrics and gynecology clinics, outpatient services, and prenatal classes.

The present study is significant as it is one of the few longitudinal studies in Korea that repeatedly measured psychological and biological changes during pregnancy. However, only the pregnancy period was included, and the study did not examine other influential factors such as past psychiatric history, postpartum period, birth outcomes, and partner (or family, spouse) factors (such as relationship, support, and psychological adjustment). In addition, the sample size is not large, which prevents the results from being statistically significant and limits the evidence of the specific symptoms and the configuration of the symptoms that constitute the maternal depressive symptoms/anxiety heterogeneity. Despite the longitudinal aspect of the models, a possible causal relationship should be considered with caution. Therefore, to find the most predictive characteristics for different depressive symptoms and anxiety trajectories, studies with larger samples and more diverse settings are suggested.

## 5. Conclusions

The present study found that depressive symptoms and anxiety in pregnant women showed several distinct trajectory groups that are associated with stress related to pregnancy. A small number of women experienced high levels of depressive symptoms throughout pregnancy, and anxiety assessed in early pregnancy persisted without abrupt changes. Perceived stress should be assessed and managed in pregnant women due to its interrelation with other mental health problems. Thus, an early screening of mental health is important in assisting healthcare providers in detecting women who are at a risk for adverse pregnancy outcomes. Additional care should be provided to identified risk groups of pregnant women, and the underlying mechanisms of those high-risk pregnant women needs to be investigated. The results of this study provide useful information for the development of interventions targeting pregnant women who may need assessment and care from healthcare providers. Further studies are needed to investigate patterns within larger and heterogeneous groups of pregnant women, including the prenatal and postnatal periods.

## Figures and Tables

**Figure 1 ijerph-18-02733-f001:**
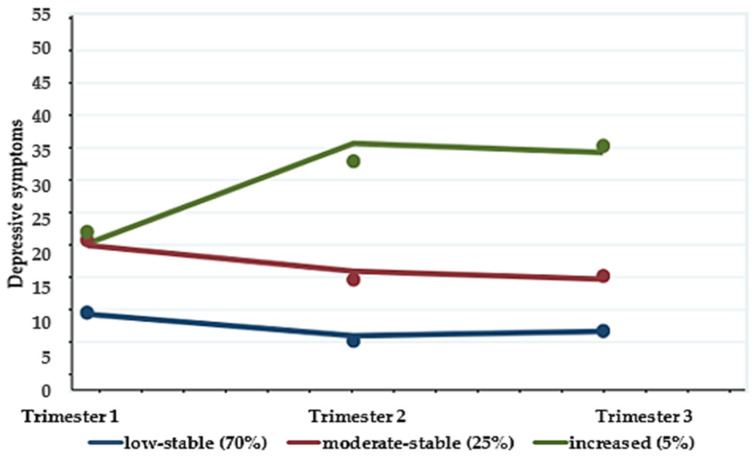
Trajectory groups for depression symptoms of pregnant women (*n* = 136).

**Figure 2 ijerph-18-02733-f002:**
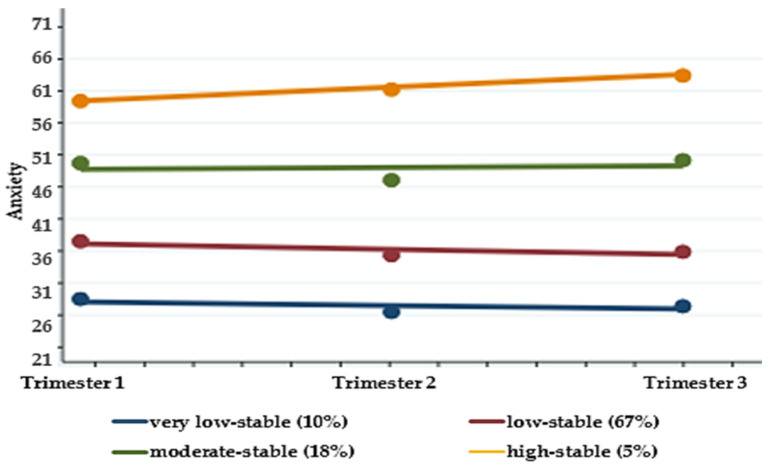
Trajectory groups for anxiety of pregnant women (*n* = 136).

**Table 1 ijerph-18-02733-t001:** Characteristics of participants (*n* = 136).

Variable		*n* (%)	Mean (SD)	Min.–Max.
Age (years)	<35	93 (68.4)		25–43
	≥35	43 (31.6)		
Education	≤High school	13 (9.6)		
	College	98 (72.1)		
	Graduate	25 (18.4)		
Employment	No	45 (36.3)		
	Yes	79 (63.7)		
Household Income	Very Low	5 (4.2)		
	Low	45 (37.8)		
	Middle	53 (44.5)		
	High	16 (13.5)		
Planned Pregnancy	No	61 (45.2)		
	Yes	74 (54.8)		
Pre-Pregnancy Smoking	No	122 (99.2)		
	Yes	1 (0.8)		
Pre-Pregnancy Drinking	No	111 (90.2)		
	Yes	12 (9.8)		
Pre-Pregnancy Regular exercise	No	100 (82.6)		
	Yes	21 (17.4)		
Pre-Pregnancy BMI	Underweight	25 (18.5)		
	Healthy weight	80 (59.3)		
	Overweight	16 (11.9)		
	Obese	14 (10.4)		
Pregnancy Stress	Trimester 1		50.71 (10.7)	28–79
	Trimester 2		68.48 (9.0)	48–87
	Trimester 3		69.02 (9.5)	44–90
Cortisol Levels	Trimester 1		4.44 (4.2)	0.1–25.6
	Trimester 2		6.23 (6.3)	0.6–49.4
	Trimester 3		7.57 (6.6)	0.3–37.6

BMI = body mass index.

**Table 2 ijerph-18-02733-t002:** Correlation between depressive symptoms, anxiety, and pregnancy stress in each trimester (*n* = 136).

Trimester	Variable	1		2		3	
		Depressive Symptoms	Anxiety	Depressive Symptoms	Anxiety	Depressive Symptoms	Anxiety
1	Anxiety	0.667 **					
	Stress	0.485 **	0.480 **				
2	Anxiety			0.750 **			
	Stress			0.479 **	0.461 **		
3	Anxiety					0.707 **	
	Stress					0.516 **	0.606 **

** *p* < 0.01.

**Table 3 ijerph-18-02733-t003:** Sociodemographic characteristics and health behaviors by trajectory groups of depressive symptoms and anxiety (*n* = 136) [*n* (%)/Mean (SD)].

Variable	Group	Depressive Symptoms Trajectory Groups				Anxiety Trajectory Groups				
		1	2	3	*p*	1	2	3	4	*p*
		(*n* = 96)	(*n* = 34)	(*n* = 6)		(*n* = 14)	(*n* = 91)	(*n* = 25)	(*n* = 6)	
Age (year)	<35	64 (66.7)	26 (76.5)	3 (50.0)	0.339	9 (64.3)	61 (67.0)	18 (72.0)	5 (83.3)	0.872
	≥35	32 (33.3)	8 (23.5)	3 (50.0)		5 (35.7)	30 (33.0)	7 (28.0)	1 (16.7)	
Education	≤High school	9 (9.4)	4 (11.8)	0 (0)	0.779	1 (7.1)	10 (11.0)	1 (4.0)	1 (16.7)	0.890
	College	67 (69.8)	26 (76.5)	5 (83.3)		11 (78.6)	65 (71.4)	18 (72.0)	4 (66.7)	
	Graduate	20 (20.8)	4 (11.8)	1 (16.7)		2 (14.3)	16 (17.6)	6 (24.0)	1 (16.7)	
Employment	No	32 (37.2)	10 (31.3)	3 (50.0)	0.675	3 (25.0)	34 (40.0)	5 (23.8)	3 (50.0)	0.391
	Yes	54 (62.8)	22 (68.7)	3 (50.0)		9 (75.0)	51 (60.0)	16 (76.2)	3 (50.0)	
Household income	Very low	4 (4.7)	1 (3.6)	0 (0)	0.913	0 (0)	4 (4.9)	1 (5.0)	0 (0)	0.883
	Low	30 (35.3)	11 (39.3)	4 (66.7)		6 (50.0)	30 (37.0)	6 (30.0)	3 (50.0)	
	Middle	39 (45.9)	12 (42.9)	2 (33.3)		6 (50.0)	36 (44.4)	9 (45.0)	2 (33.3)	
	High	12 (14.1)	4 (14.3)	0 (0)		0 (0)	11 (13.6)	4 (20.0)	1 (16.7)	
Planned pregnancy	No	39 (41.1)	17 (50.0)	5 (83.3)	0.110	6 (42.9)	43 (47.3)	9 (37.5)	3 (50.0)	0.844
	Yes	56 (58.9)	17 (50.0)	1 (16.7)		8 (57.1)	48 (52.7)	15 (62.5)	3 (50.0)	
Regular exercise	No	71 (82.6)	25 (86.2)	4 (66.7)	0.420	12 (100)	66 (80.5)	18 (85.7)	4 (66.7)	0.241
	Yes	15 (17.4)	4 (13.8)	2 (33.3)		0 (0)	16 (19.5)	3 (14.3)	2 (33.3)	
Pre-pregnancy BMI	<18.5	18 (18.9)	6 (17.6)	1 (16.7)	0.847	3 (21.4)	15 (16.5)	5 (20.8)	2 (33.3)	0.798
	18.5–23	58 (61.1)	18 (52.9)	4 (66.7)		9 (64.3)	53 (58.2)	16 (66.7)	2 (33.3)	
	23–25	10 (10.5)	6 (17.6)	0 (0)		1 (7.1)	13 (14.3)	1 (4.2)	1 (16.7)	
	25 ≤	9 (9.5)	4 (11.8)	1 (16.7)		1 (7.1)	10 (11.0)	2 (8.3)	1 (16.7)	
Pregnancy stress		47.4 (9.8)	59.3 (7.8)	57.3 (11.3)	0.000	56.9 (7.9)	50.4 (10.8)	46.4 (8.7)	58.0 (13.3)	0.017
Cortisol		4.1 (4.2)	4.9 (3.1)	7.3 (8.2)	0.171	6.8 (6.6)	4.3 (3.7)	4.1 (4.5)	2.6 (1.1)	0.189

BMI = body mass index.

**Table 4 ijerph-18-02733-t004:** Factors associated with predictors of the latent trajectory groups for depressive symptoms and anxiety.

Variables	Categories	Depressive Symptoms Trajectory Groups			Anxiety Trajectory Groups		
		OR	*p*	95% CI	OR	*p*	95% CI
Age	<35	1.179	0.755	0.42–3.32	0.603	0.273	0.24–1.49
	≥35						
Education	≤High school	1.192	0.722	0.45–3.14	1.079	0.855	0.48–2.45
	College						
	Graduate						
Employment	No	0.975	0.939	0.51–1.87	0.751	0.362	0.41–1.39
	Yes						
Household income	Very low	1.001	0.680	0.998–1.003	0.999	0.448	0.996–1.001
	Low						
	Middle						
	High						
Planned pregnancy	No	0.995	0.868	0.94–1.06	1.002	0.160	0.999–1.005
	Yes						
Pre-Pregnancy Regular exercise	No	1.303	0.674	0.38–4.47	0.646	0.405	0.23–1.81
	Yes						
Pre-Pregnancy BMI	Underweight	1.319	0.373	0.72–2.42	0.979	0.936	0.59–1.63
	Healthy weight						
	Overweight						
	Obese						
Pregnancy stress		1.133	<0.001	1.08–1.19	1.088	<0.001	1.04–1.14
Cortisol level	≤75%	0.590	0.375	0.18–1.89	0.566	0.241	0.22–1.47
	>75%						

OR = odds ration; CI = confidence interval; BMI = body mass index.

## Data Availability

The data used and/or analyzed during the current study are available from the corresponding author on request.

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
