# Peer review of "Trajectories of Depressive Symptoms and Anxiety during Pregnancy and Associations with Pregnancy Stress"

_ijerph, 2021, doi:10.3390/ijerph18052733_

Round 1
Reviewer 1 Report
The manuscript presented to me for evaluation is very interesting and carefully prepared. It takes up the subject of depression and anxiety in future mothers, which is an important topic, especially during the prevailing COVID-19 pandemic. The manuscript is generally well prepared, however, I have a few questions / comments:
- This senstence should be in different place, maybe in methods section:
"Thus, we added a saliva cortisol level measure to the psychobiological variables
of women during pregnancy to determine the difference between the distinct groups identified."
- "The results provide useful information on the development of intervention targeted for pregnant women who may need assessment and care from healthcare providers [15]"
If you want to use this sentence in introduction you shoul use future time. In this meaning this sentence should be in conclusions (not in introduction) - What are the exclusion criteria? Which women could not qualify for the study?
- When the survey was conducted, whether during the current pandemic covid-19?
- How was the research sample collected? Was random sampling used? Were all patients who consented were taken into account?
- In table 1. please insert a legend i.e.: N (%)* / Mean(SD)* and add whis abbreviation ( * or **) to each varaible.What is Range? Minimum and maximum?
- Did you calculate a minimum sample size?
- Can the authors identify which factors have the greatest impact on the trajectory of depression and anxiety? maybe using logistic regression?
Author Response
|
Reviewer’s comment |
Author’s response |
|
Reviewer 1 |
|
|
· This sentence should be in different place, maybe in methods section: "Thus, we added a saliva cortisol level measure to the psychobiological variables of women during pregnancy to determine the difference between the distinct groups identified." "The results provide useful information on the development of intervention targeted for pregnant women who may need assessment and care from healthcare providers [15]" If you want to use this sentence in introduction you should use future time. In this meaning this sentence should be in conclusions (not in introduction) |
We moved these sentences to the methods and conclusions sections; they are written in red. |
|
What are the exclusion criteria? Which women could not qualify for the study? |
We added the exclusion criteria to the methods section as follow:
Exclusion criteria were twin or triplet pregnancy and diagnosed pregnancy complications such as gestational hypertension or gestational diabetes mellitus. |
|
When the survey was conducted, whether during the current pandemic covid-19? |
This study was completed in 2016, so the study result did not reflect the COVID-19 pandemic situation. |
|
How was the research sample collected? Was random sampling used? Were all patients who consented were taken into account? |
We added the following sentences to the methods section:
This study used a repeated measures design with a convenience sampling method.
During the examination, an obstetric physician briefly introduced the study to each potential participant; a research assistant provided the detailed research protocol and obtained consent from participants who showed interest in participation in the study. |
| In table 1. please insert a legend i.e.: N (%)* / Mean(SD)* and add whis abbreviation ( * or **) to each variable. What is Range? Minimum and maximum? |
We described the n (%) and the mean (SD) in separate columns in Table 1. |
|
We changed the range to min.-max. in Table 1. |
|
| Did you calculate a minimum sample size? |
We did not calculate a minimum sample size for the study a priori. Because the trajectory analysis did not require a minimum sample size, and because the purpose of this study was to find similar patterns in mental health status among pregnant women, a sample of 136 participants was thought to be appropriate. |
| Can the authors identify which factors have the greatest impact on the trajectory of depression and anxiety? maybe using logistic regression? |
Yes, we ran the ordinal logistic regression analysis in this study and found only pregnancy stress to be a significant factor related to depression and anxiety among pregnant women. |

Reviewer 2 Report
The topic of the manuscript "Trajectories of depressive symptoms and anxiety during pregnancy and associations with pregnancy stress" is very interesting. In addition, it is of social and applied relevance. But, despite their interest, the changes mentioned below should be made to improve the quality of the manuscript and to make the results clearer.
- Introduction
The Introduction section should be revised as it does not present the research problem clearly enough and does not sufficiently summarize past evidence to make it clear what has been done and for what. On page 2, line 59 says “Although there has been extensive research on mental health during pregnancy…” Therefore, past evidence, especially on depression, should be summarized as it is hardly covered.
The purpose of the study is not entirely clear. Thus, in the Abstract, in Page 1, lines 13 and 14, it says “This study aimed to investigate trajectories of depressive symptoms and anxiety in pregnant women and the associations with pregnancy stress”. But at the end of the introductory section (page 2, lines 65-67) it says "Thus, the purposes of this study were to investigate the trajectory groups of depressive symptoms and anxiety of women during pregnancy and to identify the factors associated with those groups" . The purposes of the study should be explained more broadly, including all or the most important variables that are analyzed in the study. Could be explained what are the primary and secondary hypotheses and objectives of the study.
- Material and Methods.
The procedure for obtaining the sample and the dates on which the study was carried out should be clearly explained.
In “2.2.4. Cortisol level”, on page 3, lines 129 and 130 says “If the cortisol level was three-quarters or more of the total distribution, it was classified differently. ". It should be clearly explained how it was classified.
- Results
The title of Table 2 (page 6, line 212) “Sociodemographic characteristics by trajectory groups of depressive symptoms (n=136) [n (%) / Mean (SD)].” should be modified since this table also includes data on anxiety symptoms. Furthermore, not only sociodemographic characteristics are included.
Table 2 (pages 6 and 7) presents the characteristics by trajectory groups for depressive symptoms and anxiety. And in the text on page 6, lines 208 and 209 it says “Pregnancy stress was significantly different in the trajectory groups in both depressive symptoms and anxiety (p <.001 and p = .017, respectively). " It should be analyzed between which specific anxiety and depression groups the differences in pregnancy stress are statistically significant. Although on page 7, lines 220-222 it says “According to the proportional odds assumption, the same amount, which is 1.13 times greater, can be applied between the low-stable group and the combined categories of the medium-stable and higher groups ”, the comparison between all the groups would provide accurate information rather than being based on assumptions. And the same happens with anxiety.
Furthermore, correlational analyzes could be included between the symptoms of anxiety, depression and pregnancy stress to know the strength of the association between these variables for each trimester.
Author Response
|
Reviewer’s comment |
Revision |
|
Reviewer 2 |
|
|
We added an additional citation and revised the sentence based on the study findings (in red).
Pregnancy is a time of increased vulnerability for the development of depression. Prevalence of depression has been estimated at 11.9% among women during the perinatal period, and there is a significantly higher prevalence in low- to middle-income countries compared to high-income countries [7]. In Korea, between 20% and 40% of prenatal women have been reported to have depression, as measured by varying scales [8].
|
|
The purpose of the study is not entirely clear. Thus, in the Abstract, in Page 1, lines 13 and 14, it says “This study aimed to investigate trajectories of depressive symptoms and anxiety in pregnant women and the associations with pregnancy stress”. But at the end of the introductory section (page 2, lines 65-67) it says "Thus, the purposes of this study were to investigate the trajectory groups of depressive symptoms and anxiety of women during pregnancy and to identify the factors associated with those groups". The purposes of the study should be explained more broadly, including all or the most important variables that are analyzed in the study. Could be explained what are the primary and secondary hypotheses and objectives of the study. |
We moved the study purpose from the introduction to the abstract. In the introduction, the study purpose was described as the primary and secondary objectives. The primary objective of this study was to investigate the trajectory groups of depressive symptoms and anxiety among women during pregnancy, while the secondary objective was to identify the factors associated with those groups. |
The procedure for obtaining the sample and the dates on which the study was carried out should be clearly explained. |
We added following sentence to the methods section:
During the examination, an obstetric physician briefly introduced the study to each potential participant; a research assistant provided the detailed research protocol and obtained consent from participants who showed interest in participation in the study. |
|
In “2.2.4. Cortisol level”, on page 3, lines 129 and 130 says “If the cortisol level was three-quarters or more of the total distribution, it was classified differently. ". It should be clearly explained how it was classified. |
Cortisol levels usually show a right-skewed distribution in a healthy population. To include cortisol levels in the logistic analysis, we divided the raw data of cortisol levels into two groups by their quantiles. We revised the sentence as follows: When a cortisol level was within the lower three quantiles, it was assigned 0; when it was within the highest quantile, it was assigned 1. |
The title of Table 2 (page 6, line 212) “Sociodemographic characteristics by trajectory groups of depressive symptoms (n=136) [n (%) / Mean (SD)].” should be modified since this table also includes data on anxiety symptoms. Furthermore, not only sociodemographic characteristics are included. |
We changed the title as suggested: “Sociodemographic characteristics and health behaviors by trajectory groups of depressive symptoms and anxiety” |
|
Table 2 (pages 6 and 7) presents the characteristics by trajectory groups for depressive symptoms and anxiety. And in the text on page 6, lines 208 and 209 it says “Pregnancy stress was significantly different in the trajectory groups in both depressive symptoms and anxiety (p <.001 and p = .017, respectively). " It should be analyzed between which specific anxiety and depression groups the differences in pregnancy stress are statistically significant. |
Table 2 presented the differences in sociodemographic factors and healthy behaviors among the trajectory groups of depressive symptoms and anxiety. We added the following sentences: Compared to the low-stable depressive symptoms group, the moderate and increased groups showed significantly higher pregnancy stress (p < .001 and p = .038, respectively). The very low stable anxiety group showed significantly higher pregnancy stress than the moderate stable anxiety group (p = 0.29). |
|
Although on page 7, lines 220-222 it says “According to the proportional odds assumption, the same amount, which is 1.13 times greater, can be applied between the low-stable group and the combined categories of the medium-stable and higher groups”, the comparison between all the groups would provide accurate information rather than being based on assumptions. And the same happens with anxiety. |
We revised the sentences as follow: “Pregnancy stress significantly differed among the trajectory groups of depressive symptoms (p < 0.01). For a one-unit increase in pregnancy stress, the odds of a higher category group are 1.13 times greater in depressive symptoms (for example, the low-stable group [group 1] versus the moderate-stable group [group 2]) when other variables in the model are held constant. The same amount, which is 1.13 times greater for depressive symptoms, can be also applied between the moderate-stable and increased groups.” “In terms of anxiety, pregnancy stress significantly differed among the trajectory groups of anxiety (p < 0.01). For a one-unit increase in pregnancy stress, the odds of a higher category group are 1.09 times greater in anxiety (for example, the very-low stable group [group 1] versus the low stable group [group 2]) when the other variables in the model remain constant. The same increase, 1.09 times, can be applied between the low-stable group (group 2) and the moderate-stable group (group 3).” |
|
Furthermore, correlational analyzes could be included between the symptoms of anxiety, depression and pregnancy stress to know the strength of the association between these variables for each trimester. |
The result of the ordinal logistic regression presents the association of pregnancy stress with depression and anxiety by p-value, and this study was conducted to identify the groups with similar patterns of depressive symptoms and anxiety. We thought it best not to include the descriptions of each trimester. |

Reviewer 3 Report
The manuscript titled "Trajectories of depressive sysmptoms and anxiety during pregnancy and associations with pregnancy stress" investigates the association between pregnancy anxiety and stress.
Introduction:
"Mental health statuses such as depression, anxiety, and stress, are commonly reported in pregnant women, as well as in general healthy populations" - this just means that independent of being pregnant those statses are common. So pelase, do not bias the information about pregnancy.
"Anxiety is another prevalent mental status faced by pregnant women, and is often reported with depression" - does that mean, that pregnant anxiety is associated with depression? If yes, pelase present those values or directly indicate this in the sentence. If not, please clearly signalize their non-association.
What about a revision on anxiety and depression in those women, who already suffer from this before pregancy? What are the comparisons between them? Does the literature review on this topic?
Materials and methods:
"To reduce variations in measurements, one person measured all parameters throughout the study" - just one person analyzed the samples? What if this one person makes a mistake, what about the consequences on not having a backup analyst to reprove the samples?
"Age was categorized as < 35 or ≥ 35 years" why categorize age in only 2 groups? This could bias the sample information and analysis. Almost 70% of participants are under 35 years. What about this age distribution? Is such an unbalanced categorization.
Results:
Figures 1 and 2: please improve figure quality. Legend is not clearly readable.
What about a combined trajectory group for both depression and anxiety?
Pregnancy stress was statsitically significant in the between groups analysis for both depressive and anxiety symptoms. Where are those differences? Since there are more than 2 groups being compared, a pairwise group comparison should be performed to investigate where the differences come from.
Discussion:
A clear linl between depression and anxiety trajectories is missing.
It seems that authors look for a valdiation of their findings, regarding both trajectory groups, compared to the literature. What are the noveltys that they can deliver the readers?
Author Response
|
Reviewer’s comment |
Revision |
|
Reviewer 3 |
|
|
Introduction: "Mental health statuses such as depression, anxiety, and stress, are commonly reported in pregnant women, as well as in general healthy populations" - this just means that independent of being pregnant those states are common. So please, do not bias the information about pregnancy. |
We revised the sentence as follows: “Mental health problems such as depression and anxiety are prevalent among pregnant women, with rates as high as 30%.” |
|
"Anxiety is another prevalent mental status faced by pregnant women, and is often reported with depression" - does that mean, that pregnant anxiety is associated with depression? If yes, please present those values or directly indicate this in the sentence. If not, please clearly signalize their non-association. |
We revised the sentence as follows and added a supporting citation: “Anxiety is another prevalent problem faced by pregnant women [1,9].” “A strong relationship between anxiety and depression has been established among pregnant women [12]. Three out of every four anxious women are reported to suffer from comorbid depressive symptoms.” |
|
What about a revision on anxiety and depression in those women, who already suffer from this before pregnancy? What are the comparisons between them? Does the literature review on this topic? |
We added the following sentence to the introduction:
“Various factors influencing prenatal anxiety and depression in women have been investigated; however, inconsistent reports are available regarding stress, sleep quality, and daily physical activity. Mental health problems before pregnancy, on the other hand, have been identified as strong predictors of perinatal and postnatal mental health problems [14-17].” |
|
Materials and methods: "To reduce variations in measurements, one person measured all parameters throughout the study" - just one person analyzed the samples? What if this one person makes a mistake, what about the consequences on not having a backup analyst to reprove the samples? |
We deleted the following sentence to avoid unnecessary confusion for readers: “To reduce variations in measurements, one person measured all parameters throughout the study.” This sentence was originally written to explain the procedures for handling the saliva samples and equipment. The intra- and inter-assay coefficients of variation for salivary cortisol reported in this study were in ranges recommended as < 10 for intra-assay and 15% for inter-assay coefficients of variation. |
|
"Age was categorized as < 35 or ≥ 35 years" why categorize age in only 2 groups? This could bias the sample information and analysis. Almost 70% of participants are under 35 years. What about this age distribution? Is such an unbalanced categorization. |
According to the WHO’s definition, pregnant women over 35 years old are considered high-risk for pregnancy complications such as preterm delivery, hypertension, superimposed preeclampsia, and severe preeclampsia (Cavazos-Rehg et al., 2015). We added the range of participants’ ages to Table 1. |
|
We improved the figures so that the reader can understand them more easily. |
|
|
We were interested in finding similar patterns of depressive symptoms and anxiety separately, but they are closely related. Thus, we did not have to analyze the combined variables of both mental statuses. |
|
|
Pregnancy stress was statistically significant in the between groups analysis for both depressive and anxiety symptoms. Where are those differences? Since there are more than 2 groups being compared, a pairwise group comparison should be performed to investigate where the differences come from. |
We added following sentences: “Compared to the low-stable depressive symptoms group, the moderate and increased groups showed significantly higher pregnancy stress (p < .001 and p = .038, respectively). The very low stable anxiety group showed significantly higher pregnancy stress than the moderate stable anxiety group (p = 0.29).” |
|
Discussion: A clear linl between depression and anxiety trajectories is missing. |
“The major finding of this study indicates that a small number of women showed increased depressive symptoms as their pregnancies progressed; therefore, greater attention and care are needed to prevent adverse obstetric outcomes within this group of women. Unlike depressive symptoms, anxiety assessed during early pregnancy did not change abruptly, so early and timely screening by healthcare providers could reduce anxiety symptoms in pregnant women.” |
|
This study has two purposes: the identification of the trajectories of depressive symptoms and anxiety and the determination of influencing factors within the trajectory groups. Depression and anxiety in pregnant women have been of interest to researchers due to the negative consequences on pregnancy and, thus, need to be investigated in different populations and settings. This study also included an objective measure; salivary cortisol was treated as a potential factor but was not found to be a significant factor. Future study investigating the role of cortisol in depression and anxiety in pregnant women is warranted. |
|

Round 2
Reviewer 1 Report
The authors of the presented manuscript addressed all the comments of the reviewer. They explained the disputable issues or made appropriate corrections and comments. I believe that the article deserves to be published in a IJERPH.
Author Response
Thank you very much for your comments.
Reviewer 2 Report
The revised manuscript " Trajectories of depressive symptoms and anxiety during pregnancy and associations with pregnancy stress " has improved compared to the initial version, however some problems and deficiencies persist.
Page 7, lines 432-433 says "low-stable anxiety group showed significantly higher pregnancy stress than the moderate-stable anxiety group (p = 0.29)." If p = 0.29 the differences are not statistically significant.
In the initial review it was stated:
"Furthermore, correlational analyzes could be included between the symptoms of anxiety, depression and pregnancy stress to know the strength of the association between these variables for each trimester."
The authors answered this:
“The result of the ordinal logistic regression presents the association of pregnancy stress with depression and anxiety by p-value, and this study was conducted to identify the groups with similar patterns of depressive symptoms and anxiety. We thought it best not to include the descriptions of each trimester."
The reason why they do not want to put the description of each trimester is not clear, in any case the extent to which anxiety is associated with depression in the present work should be clearly presented in the Results section. The importance of the correlation between anxiety and depression is recognized in the Introduction section. So, on page 2, lines 91-93 it says “A strong relationship between anxiety and depression has been established among pregnant women [12]. Three out of every four anxious women are reported to suffer from comorbid depressive symptoms." And on line 96 it says Line 96 "Given the high correlation of anxiety and depression,". Therefore, it is not understood that the results found in this study on the association between anxiety and depression are not treated explicitly, clearly and completely.
Author Response
|
Reviewer’s comment |
Revision |
|
Reviewer 2 |
|
|
Thank you for your comment. We corrected the p-value to 0.029 in the manuscript. (P7, L237) |
|
|
In the initial review it was stated: "Furthermore, correlational analyzes could be included between the symptoms of anxiety, depression and pregnancy stress to know the strength of the association between these variables for each trimester." The authors answered this: “The result of the ordinal logistic regression presents the association of pregnancy stress with depression and anxiety by p-value, and this study was conducted to identify the groups with similar patterns of depressive symptoms and anxiety. We thought it best not to include the descriptions of each trimester." The reason why they do not want to put the description of each trimester is not clear, in any case the extent to which anxiety is associated with depression in the present work should be clearly presented in the Results section. The importance of the correlation between anxiety and depression is recognized in the Introduction section. So, on page 2, lines 91-93 it says “A strong relationship between anxiety and depression has been established among pregnant women [12]. Three out of every four anxious women are reported to suffer from comorbid depressive symptoms." And on line 96 it says Line 96 "Given the high correlation of anxiety and depression,". Therefore, it is not understood that the results found in this study on the association between anxiety and depression are not treated explicitly, clearly and completely. |
Thank you for your comments. We added the table and interpretation regarding the correlation between depressive symptoms, anxiety, and pregnancy stress in each trimester. Associations between depressive symptoms, anxiety, and pregnancy-related stress of pregnant women are statistically significant in each trimester. All p-values of correlation are smaller than 0.001, as illustrated in Table 2. In all trimesters, depressive symptoms and stress were highly correlated, and anxiety and stress were also significantly correlated. For the next step, we conducted the ordinal logistic regression to identify the factors affecting the trajectory groups of depressive symptoms and anxiety of women in pregnancy. |

Reviewer 3 Report
The comments and poitings from all reviewers were adressed and significant improvements were performed.
Author Response
Thank you for your comment.